# Degree-Quant: Quantization-Aware Training for Graph Neural Networks

**Shyam A. Tailor**[*]
Department of Computer Science & Technology
University of Cambridge

**Javier Fernandez-Marques**[*]
Department of Computer Science
University of Oxford

**Nicholas D. Lane**
Department of Computer Science and Technology
University of Cambridge
& Samsung AI Center

## Abstract

Graph neural networks (GNNs) have demonstrated strong performance on a wide variety of tasks due to their ability to model non-uniform structured data. Despite their promise, there exists little research exploring methods to make them more efficient at inference time. In this work, we explore the viability of training quantized GNNs, enabling the usage of low precision integer arithmetic during inference. For GNNs seemingly unimportant choices in quantization implementation cause dramatic changes in performance. We identify the sources of error that uniquely arise when attempting to quantize GNNs, and propose an architecturally-agnostic and stable method, *Degree-Quant*, to improve performance over existing quantization-aware training baselines commonly used on other architectures, such as CNNs. We validate our method on six datasets and show, unlike previous attempts, that models generalize to unseen graphs. Models trained with Degree-Quant for INT8 quantization perform as well as FP32 models in most cases; for INT4 models, we obtain up to 26% gains over the baselines. Our work enables up to $4.7\times$ speedups on CPU when using INT8 arithmetic.

## 1 Introduction

GNNs have received substantial attention in recent years due to their ability to model irregularly structured data. As a result, they are extensively used for applications as diverse as molecular interactions (Duvenaud et al., 2015; Wu et al., 2017), social networks (Hamilton et al., 2017), recommendation systems (van den Berg et al., 2017) or program understanding (Allamanis et al., 2018). Recent advancements have centered around building more sophisticated models including new types of layers (Kipf & Welling, 2017; Velickovic et al., 2018; Xu et al., 2019) and better aggregation functions (Corso et al., 2020). However, despite GNNs having few model parameters, the compute required for each application remains tightly coupled to the input graph size. A 2-layer Graph Convolutional Network (GCN) model with 32 hidden units would result in a model size of just 81KB but requires 19 GigaOPs to process the entire Reddit graph. We illustrate this growth in fig. 1.

One major challenge with graph architectures is therefore performing inference efficiently, which limits the applications they can be deployed for. For example, GNNs have been combined with CNNs for SLAM feature matching (Sarlin et al., 2019), however it is not trivial to deploy this technique on smartphones, or even smaller devices, whose neural network accelerators often do not implement floating point arithmetic, and instead favour more efficient integer arithmetic. Integer quantization is one way to lower the compute, memory and energy budget required to perform inference, without requiring modifications to the model architecture; this is also useful for model serving in data centers.

Although quantization has been well studied for CNNs and language models (Jacob et al., 2017; Wang et al., 2018; Zafrir et al., 2019; Prato et al., 2019), there remains relatively little work addressing

---

[*]Equal contribution. Correspondence to: Shyam Tailor <sat62@cam.ac.uk>

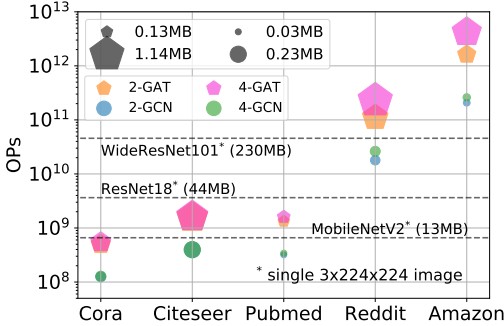

**Figure 1:** Despite GNN model sizes rarely exceeding 1MB, the OPs needed for inference grows at least linearly with the size of the dataset and node features. GNNs with models sizes $100\times$ smaller than popular CNNs require many more OPs to process large graphs.

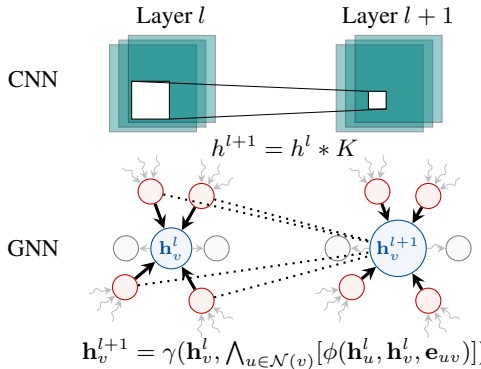

$$\mathbf{h}_v^{l+1} = \gamma(\mathbf{h}_v^l, \textstyle\bigwedge_{u \in \mathcal{N}(v)} [\phi(\mathbf{h}_u^l, \mathbf{h}_v^l, \mathbf{e}_{uv})])$$

**Figure 2:** While CNNs operate on regular grids, GNNs operate on graphs with varying topology. A node's neighborhood size and ordering varies for GNNs. Both architectures use weight sharing.

GNN efficiency (Mukkara et al., 2018; Jia et al., 2020; Zeng & Prasanna, 2020; Yan et al., 2020). To the best of our knowledge, there is no work explicitly characterising the issues that arise when quantizing GNNs or showing latency benefits of using low-precision arithmetic. The recent work of Wang et al. (2020) explores only binarized embeddings of a single graph type (citation networks). In Feng et al. (2020) a heterogeneous quantization framework assigns different bits to embedding and attention coefficients in each layer while maintaining the weights at full precision (FP32). Due to the mismatch in operands' bit-width the majority of the operations are performed at FP32 after data casting, making it impractical to use in general purpose hardware such as CPUs or GPUs. In addition they do not demonstrate how to train networks which generalize to *unseen* input graphs. Our framework relies upon uniform quantization applied to all elements in the network and uses bit-widths (8-bit and 4-bit) that are supported by off-the-shelf hardware such as CPUs and GPU for which efficient low-level operators for common operations found in GNNs exists.

This work considers the motivations and problems associated with quantization of graph architectures, and provides the following contributions:

- The explanation of the sources of degradation in GNNs when using lower precision arithmetic. We show how the choice of straight-through estimator (STE) implementation, node degree, and method for tracking quantization statistics significantly impacts performance.

- An *architecture-agnostic* method for quantization-aware training on graphs, *Degree-Quant* (DQ), which results in INT8 models often performing as well as their FP32 counterparts. At INT4, models trained with DQ typically outperform quantized baselines by over 20%. We show, unlike previous work, that models trained with DQ generalize to *unseen graphs*. We provide code at this URL: `https://github.com/camlsys/degree-quant`.

- We show that quantized networks achieve up to $4.7\times$ speedups on CPU with INT8 arithmetic, relative to full precision floating point, with $4\text{-}8\times$ reductions in runtime memory usage.

## 2 BACKGROUND

### 2.1 MESSAGE PASSING NEURAL NETWORKS (MPNNS)

Many popular GNN architectures may be viewed as generalizations of CNN architectures to an irregular domain: at a high level, graph architectures attempt to build representations based on a node's neighborhood (see fig. 2). Unlike CNNs, however, this neighborhood does not have a fixed ordering or size. This work considers GNN architectures conforming to the MPNN paradigm (Gilmer et al., 2017). A graph $\mathcal{G} = (V, E)$ has node features $\mathbf{X} \in \mathbb{R}^{N \times F}$, an incidence matrix $\mathbf{I} \in \mathbb{N}^{2 \times E}$, and optionally $D$-dimensional edge features $\mathbf{E} \in \mathbb{R}^{E \times D}$. The forward pass through an MPNN layer consists of message passing, aggregation and update phases: $\mathbf{h}_{l+1}^{(i)} = \gamma(\mathbf{h}_l^{(i)}, \bigwedge_{j \in \mathcal{N}(i)} [\phi(\mathbf{h}_l^{(j)}, \mathbf{h}_l^{(i)}, \mathbf{e}_{ij})])$. Messages from

node $u$ to node $v$ are calculated using function $\phi$, and are aggregated using a permutation-invariant function $\bigwedge$. The features at $v$ are subsequently updated using $\gamma$.

We focus on three architectures with corresponding update rules:

1. Graph Convolution Network (GCN): $\mathbf{h}_{l+1}^{(i)} = \sum_{j \in \mathcal{N}(i) \cup \{i\}} \left( \frac{1}{\sqrt{d_i d_j}} \mathbf{W} \mathbf{h}_l^{(j)} \right)$ (Kipf & Welling, 2017), where $d_i$ refers to the degree of node $i$.

2. Graph Attention Network (GAT): $\mathbf{h}_{l+1}^{(i)} = \alpha_{i,i} \mathbf{W} \mathbf{h}_l^{(i)} + \sum_{j \in \mathcal{N}(i)} (\alpha_{i,j} \mathbf{W} \mathbf{h}_l^{(j)})$, where $\alpha$ represent attention coefficients (Velickovic et al., 2018).

3. Graph Isomorphism Network (GIN): $\mathbf{h}_{l+1}^{(i)} = f_{\boldsymbol{\Theta}}[(1 + \epsilon)\mathbf{h}_l^{(i)} + \sum_{j \in \mathcal{N}(i)} \mathbf{h}_l^{(j)}]$, where $f$ is a learnable function (e.g. a MLP) and $\epsilon$ is a learnable constant (Xu et al., 2019).

## 2.2 QUANTIZATION FOR NON-GRAPH NEURAL NETWORKS

Quantization allows for model size reduction and inference speedup without changing the model architecture. While there exists extensive studies of the impact of quantization at different bit-widths (Courbariaux et al., 2015; Han et al., 2015; Louizos et al., 2017) and data formats (Micikevicius et al., 2017; Carmichael et al., 2018; Kalamkar et al., 2019), it is 8-bit integer (INT8) quantization that has attracted the most attention. This is due to INT8 models reaching comparable accuracy levels to FP32 models (Krishnamoorthi, 2018; Jacob et al., 2017), offer a $4\times$ model compression, and result in inference speedups on off-the-shelf hardware as 8-bit arithmetic is widely supported.

Quantization-aware training (QAT) has become the *de facto* approach towards designing robust quantized models with low error (Wang et al., 2018; Zafrir et al., 2019; Wang et al., 2018). In their simplest forms, QAT schemes involve exposing the numerical errors introduced by quantization by simulating it on the forward pass Jacob et al. (2017) and make use of STE (Bengio et al., 2013) to compute the gradients—as if no quantization had been applied. For integer QAT, the quantization of a tensor $x$ during the forward pass is often implemented as: $x_q = \min(q_{\max}, \max(q_{\min}, \lfloor x/s + z \rfloor))$, where $q_{\min}$ and $q_{\max}$ are the minimum and maximum representable values at a given bit-width and signedness, $s$ is the scaling factor making $x$ span the $[q_{\min}, q_{\max}]$ range and, $z$ is the *zero-point*, which allows for the real value 0 to be representable in $x_q$. Both $s$ and $z$ are scalars obtained at training time. hen, the tensor is *dequantized* as: $\hat{x} = (x_q - z)s$, where the resulting tensor $\hat{x} \sim x$ for a high enough bit-width. This similarity degrades at lower bit-widths. Other variants of integer QAT are presented in Jacob et al. (2017) and Krishnamoorthi (2018).

To reach performance comparable to FP32 models, QAT schemes often rely on other techniques such as *gradient clipping*, to mask gradient updates based on the largest representable value at a given bit-width; stochastic, or noisy, QAT, which stochastically applies QAT to a portion of the weights at each training step (Fan et al., 2020; Dong et al., 2017); or the re-ordering of layers (Sheng et al., 2018; Alizadeh et al., 2019).

## 3 QUANTIZATION FOR GNNS

In this section, we build an intuition for why GNNs would fail with low precision arithmetic by identifying the sources of error that will disproportionately affect the accuracy of a low precision model. Using this insight, we propose our technique for QAT with GNNs, *Degree-Quant*. Our analysis focuses on three models: GCN, GAT and GIN. This choice was made as we believe that these are among the most popular graph architectures, with strong performance on a variety of tasks (Dwivedi et al., 2020), while also being representative of different trends in the literature.

## 3.1 SOURCES OF ERROR

QAT relies upon the STE to make an estimate of the gradient despite the non-differentiable rounding operation in the forward pass. If this approximation is inaccurate, however, then poor performance will be obtained. In GNN layers, we identify the aggregation phase, where nodes combine messages from a varying number of neighbors in a permutation-invariant fashion, as a source of substantial numerical error, especially at nodes with high in-degree. Outputs from aggregation have magnitudes

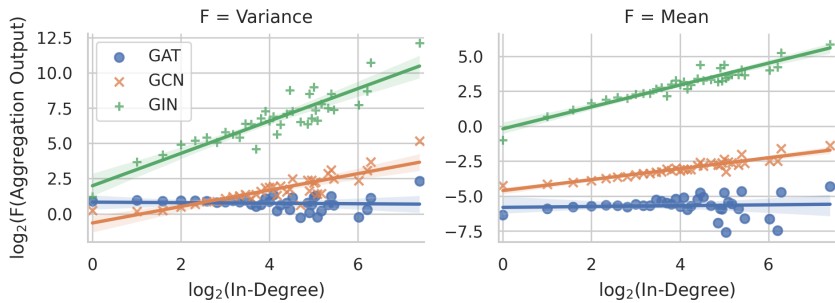

**Figure 3:** Analysis of values collected immediately after aggregation at the final layer of FP32 GNNs trained on Cora. Generated using channel data collected from 100 runs for each architecture. As in-degree grows, so does the mean and variance of channel values after aggregation.

that vary significantly depending on a node's in-degree: as it increases, the variance of aggregation values will increase.[1] Over the course of training $q_{\min}$ and $q_{\max}$, the quantization range statistics, become severely distorted by infrequent outliers, reducing the resolution for the vast majority of values observed. This reults in increased rounding error for nodes with smaller in-degrees. Controlling $q_{\min}$ and $q_{\max}$ hence becomes a trade-off balancing *truncation error* and *rounding error*.

We can derive how the mean and variance of the aggregation output values vary as node in-degree, $n$, increases for each of the three GNN layers. Suppose we model incoming message values for a single output dimension with random variables $X_i$, without making assumptions on their exact distribution or independence. Further, we use $Y_n$ as the random variable representing the value of node output after the aggregation step. With GIN layers, we have $Y_n = (1 + \epsilon)X_0 + \sum_{i=1}^{n} X_i$. It is trivial to prove that $\mathbb{E}(Y_n) = \mathcal{O}(n)$. The variance of the aggregation output is also $\mathcal{O}(n)$ in the case that that $\sum_{i \neq j} \text{Cov}(X_i, X_j) \ll \sum_i \text{Var}(X_i)$. We note that if $\sum_{i \neq j} \text{Cov}(X_i, X_j)$ is large then it implies that the network has learned highly redundant features, and may be a sign of over-fitting. Similar arguments can be made for GCN and GAT layers; we would expect GCN aggregation values to grow like $\mathcal{O}(\sqrt{n})$, and GAT aggregation values to remain constant ($\mathcal{O}(1)$) due to the attention coefficients.

We empirically validate these predictions on GNNs trained on Cora; results are plotted in fig. 3. We see that the aggregation values do follow the trends predicted, and that for the values of in-degree in the plot (up to 168) the covariance terms can be neglected. As expected, the variance and mean of the aggregated output grow fastest for GIN, and are roughly constant for GAT as in-degree increases. From this empirical evidence, it would be expected that GIN layers are most affected by quantization.

By using GIN and GCN as examples, we can see how aggregation error causes error in weight updates. Suppose we consider a GIN layer incorporating one weight matrix in the update function i.e. $\mathbf{h}_{l+1}^{(i)} = f(\mathbf{W}\mathbf{y}_{\text{GIN}}^{(i)})$, where $f$ is an activation function, $\mathbf{y}_{\text{GIN}}^{(i)} = (1 + \epsilon)\mathbf{h}_l^{(i)} + \sum_{j \in \mathcal{N}(i)} \mathbf{h}_l^{(j)}$, and $\mathcal{N}(i)$ denotes the in-neighbors of node $i$. Writing $\mathbf{y}_{\text{GCN}}^{(i)} = \sum_{k \in \mathcal{N}(i)} (\frac{1}{\sqrt{d_i d_k}} \mathbf{W} \mathbf{h}_l^{(j)})$, we see that the derivatives of the loss with respect to the weights for GCN and GIN are:

$$
\begin{array}{cc}
\textbf{GIN} & \textbf{GCN} \\
\frac{\partial \mathcal{L}}{\partial \mathbf{W}} = \sum_{i=1}^{|V|} \left( \frac{\partial \mathcal{L}}{\partial \mathbf{h}_{l+1}^{(i)}} \circ f'(\mathbf{W}\mathbf{y}_{\text{GIN}}^{(i)}) \right) \mathbf{y}_{\text{GIN}}^{(i)\top} & \frac{\partial \mathcal{L}}{\partial \mathbf{W}} = \sum_{i=1}^{|V|} \sum_{j \in \mathcal{N}(i)} \frac{1}{\sqrt{d_i d_j}} \left( \frac{\partial \mathcal{L}}{\partial \mathbf{h}_{l+1}^{(i)}} \circ f'(\mathbf{y}_{\text{GCN}}^{(i)}) \right) \mathbf{h}_l^{(j)\top}
\end{array}
$$

The larger the error in $\mathbf{y}_{\text{GIN}}^{(i)}$—caused by aggregation error—the greater the error in the weight gradients for GIN, which results in poorly performing models being obtained. The same argument applies to GCN, with the $\mathbf{h}_l^{(j)\top}$ and $\mathbf{y}_{\text{GCN}}^{(i)}$ terms introducing aggregation error into the weight updates.

### 3.2 OUR METHOD: DEGREE-QUANT

To address these sources of error we propose *Degree-Quant* (DQ), a method for QAT with GNNs. We consider both *inaccurate weight updates* and *unrepresentative quantization ranges*.

---

[1] The reader should note that we are not referring to the concept of estimator variance, which is the subject of sampling based approaches—we are exclusively discussing the variance of values immediately after aggregation.

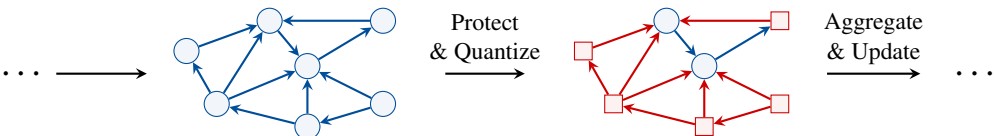

**Figure 4:** High-level view of the stochastic element of Degree-Quant. Protected (high in-degree) nodes, in blue, operate at full precision, while unprotected nodes (red) operate at reduced precision. High in-degree nodes contribute most to poor gradient estimates, hence they are stochastically protected from quantization more often.

---

**Algorithm 1** Degree-Quant (DQ). Functions accepting a protective mask $\mathbf{m}$ perform only the masked computations at full precision: intermediate tensors are *not* quantized. At test time protective masking is disabled. In fig. 11 (in the Appendix) we show with a diagram how a GCN layers makes use of DQ.

---

1: **procedure** TRAINFORWARDPASS($\mathcal{G}, \mathbf{p}$)
2:     ▷ Calculate mask and *quantized* weights, $\Theta'$, which all operations share
3:     $\mathbf{m} \leftarrow$ BERNOULLI($\mathbf{p}$)
4:     $\Theta' \leftarrow$ QUANTIZE($\Theta$)
5:     ▷ Messages with masked sources are at full precision (excluding weights)
6:     $\mathcal{M} \leftarrow$ MESSAGECALCULATE($\mathcal{G}, \Theta', \mathbf{m}$)
7:     $X \leftarrow$ QUANTIZE(AGGREGATE($\mathcal{M}, \Theta', \mathbf{m}$), $\mathbf{m}$)          ▷ No quantization for masked nodes
8:     **return** UPDATE($X, \Theta', \mathbf{m}$)          ▷ Quantized weights always used
9: **end procedure**

---

**Stochastic Protection from Quantization to Improve Weight Update Accuracy**. DQ aims to encourage more accurate weight updates by stochastically protecting nodes in the network from quantization. At each layer a protective node mask is generated; all masked nodes have the phases of the message passing, aggregation and update performed at full precision. This includes messages sent by protected nodes to other nodes, as shown in fig. 4 (a detailed diagram is shown in fig. 11). It is also important to note that the weights used at all nodes are the same quantized weights; this is motivated by the fact that our method is used to encourage more accurate gradients to flow back to the weights through high in-degree nodes. At test time protection is disabled: all nodes operate at low precision.

To generate the mask, we pre-process each graph before training and create a vector of probabilities $\mathbf{p}$ with length equal to the number of nodes. At training time, mask $\mathbf{m}$ is generated by sampling using the Bernoulli distribution: $\mathbf{m} \sim$ Bernoulli($\mathbf{p}$). In our scheme $p_i$ is higher if the in-degree of node $i$ is large, as we find empirically that high in-degree nodes contribute most towards error in weight updates. We use a scheme with two hyperparameters, $p_{\min}$ and $p_{\max}$; nodes with the maximum in-degree are assigned $p_{\max}$ as their masking probability, with all other nodes assigned a probability calculated by interpolating between $p_{\min}$ and $p_{\max}$ based on their in-degree ranking in the graph.

**Percentile Tracking of Quantization Ranges**. Figure 3 demonstrates large fluctuations in the variance of the aggregation output as in-degree increases. Since these can disproportionately affect the ranges found by using min-max or momentum-based quantization, we propose using *percentiles*. While percentiles have been used for post-training quantization (Wu et al., 2020), we are the first (to the best of our knowledge) to propose making it a core part of QAT; we find it to be a key contributor to achieving consistent results with graphs. Using percentiles involves ordering the values in the tensor and clipping a fraction of the values at both ends of the distribution. The fraction to clip is a hyperparameter. We are more aggressive than existing literature on the quantity we discard: we clip the top and bottom 0.1%, rather than 0.01%, as we observe the fluctuations to be a larger issue with GNNs than with CNNs or DNNs. Quantization ranges are more representative of the vast majority of values in this scheme, resulting in less *rounding error*.

We emphasize that a core contribution of DQ is that it is *architecture-agnostic*. Our method enables a wide variety of architectures to use low precision arithmetic at inference time. Our method is also *orthogonal*—and complementary—to other techniques for decreasing GNN computation requirements, such as sampling based methods which are used to reduce memory consumption (Zeng et al., 2020), or weight pruning (Blalock et al., 2020) approaches to achieve further model compression.

| Dataset | Model Arch. | *vanilla* STE | | | | STE with Gradient Clipping | | | |
| --- | --- | --- | --- | --- | --- | --- | --- | --- | --- |
| | | min/max | | momentum | | min/max | | momentum | |
| | | W8A8 | W4A4 | W8A8 | W4A4 | W8A8 | W4A4 | W8A8 | W4A4 |
| Cora (Acc. %)↑ | GCN | **81.0 ± 0.7** | 65.3 ± 4.9 | 42.3 ± 11.1 | 49.4 ± 8.8 | 80.8 ± 0.8 | 62.3 ± 5.2 | 66.9 ± 18.2 | **77.2 ± 2.5** |
| | GAT | 76.0 ± 2.2 | 16.8 ± 8.5 | 81.7 ± 1.3 | **51.7 ± 5.8** | 76.4 ± 2.6 | 15.4 ± 8.1 | **81.9 ± 0.7** | 47.4 ± 5.0 |
| | GIN | 69.9 ± 1.9 | 25.9 ± 2.6 | 49.2 ± 10.2 | 42.8 ± 4.0 | 69.2 ± 2.3 | 29.5 ± 3.5 | **75.1 ± 1.1** | 40.5 ± 5.0 |
| MNIST (Acc. %)↑ | GCN | **90.4 ± 0.2** | 51.3 ± 7.5 | 90.1 ± 0.5 | **70.6 ± 2.4** | 90.4 ± 0.3 | 54.8 ± 1.5 | 90.2 ± 0.4 | 10.3 ± 0.0 |
| | GAT | **95.8 ± 0.1** | 20.1 ± 3.3 | 95.7 ± 0.3 | 67.4 ± 3.2 | 95.7 ± 0.1 | 30.2 ± 7.4 | 95.7 ± 0.3 | **76.3 ± 1.2** |
| | GIN | 96.5 ± 0.3 | 62.4 ± 21.8 | **96.7 ± 0.2** | 91.0 ± 0.6 | 96.4 ± 0.4 | 19.5 ± 2.1 | 75.3 ± 18.1 | 10.8 ± 0.9 |
| ZINC (Loss)↓ | GCN | 0.486 ± 0.01 | 0.747 ± 0.02 | 0.509 ± 0.01 | 0.710 ± 0.05 | 0.495 ± 0.01 | 0.766 ± 0.02 | **0.483 ± 0.01** | **0.692 ± 0.01** |
| | GAT | 0.471 ± 0.01 | 0.740 ± 0.02 | 0.571 ± 0.03 | **0.692 ± 0.06** | 0.466 ± 0.01 | 0.759 ± 0.04 | **0.463 ± 0.01** | 0.717 ± 0.03 |
| | GIN | 0.393 ± 0.02 | 1.206 ± 0.27 | **0.386 ± 0.03** | **0.572 ± 0.02** | 0.390 ± 0.02 | 1.669 ± 0.10 | 0.388 ± 0.02 | 0.973 ± 0.24 |

**Table 1:** Impact on performance of four typical quantization implementations for INT8 and INT4. The configuration that resulted in best performing models for each dataset-model pair is bolded. Hyperparameters for each experiment were fine-tuned independently. As expected, adding clipping does not change performance with min/max but does with momentum. **A major contribution of this work is identifying that seemingly unimportant choices in quantization implementation cause dramatic changes in performance.**

## 4 EXPERIMENTS

In this section we first analyse how the choice of quantization implementation affects performance of GNNs. We subsequently evaluate Degree-Quant against the strong baselines of: FP32, INT8-QAT and, INT8-QAT with stochastic masking of weights (Fan et al., 2020). We refer to this last approach as *noisy* QAT or nQAT. To make explicit that we are quantizing both weights and activations, we use the notation W8A8. We repeat the experiments at INT4. Our study evaluates performance on six datasets and includes both node-level and graph-level tasks. The datasets used were Cora, CiteSeer, ZINC, MNIST and CIFAR10 superpixels, and REDDIT-BINARY. Across all datasets INT8 models trained with Degree-Quant manage to recover most of the accuracy lost as a result of quantization. In some instances, DQ-INT8 outperform the extensively tuned FP32 baselines. For INT4, DQ outperforms all QAT baselines and results in double digits improvements over QAT-INT4 in some settings. Details about each dataset and our experimental setup can be found in appendix A.1.

### 4.1 IMPACT OF QUANTIZATION GRADIENT ESTIMATOR ON CONVERGENCE

The STE is a workaround for when the forward pass contains non-differentiable operations (e.g. rounding in QAT) that has been widely adopted in practice. While the choice of STE implementation generally results in marginal differences for CNNs—even for binary networks (Alizadeh et al., 2019)—it is unclear whether only marginal differences will also be observed for GNNs. Motivated by this, we study the impact of four off-the-shelve quantization procedures on the three architectures evaluated for each type of dataset; the implementation details of each one is described in appendix A.3. We perform this experiment to ensure that we have the strongest possible QAT baselines. Results are shown in table 1. We found the choice quantization implementation to be highly dependent on the model architecture and type of problem to be solved: we see a much larger variance than is observed with CNNs; this is an important discovery for future work building on our study.

We observe a general trend in all INT4 experiments benefiting from momentum as it helps smoothing out the quantization statistics for the inherently noisy training stage at low bitwidths. This trend applies as well for the majority of INT8 experiments, while exhibiting little impact on MNIST. For INT8 Cora-GCN, large gradient norm values in the early stages of training (see fig. 5) mean that these models not benefit from momentum as quantization ranges fail to keep up with the rate of changes in tensor values; higher momentum can help but also leads to instability. In contrast, GAT has stable initial training dynamics, and hence obtains better results with momentum. For the molecules dataset ZINC, we consistently obtained lower regression loss when using momentum. We note that GIN models often suffer from higher performance degradation (as was first noted in fig. 3), specially at W4A4. This is not the case however for image datasets using superpixels. We believe that datasets with Gaussian-like node degree distributions (see fig. 9) are more tolerant of the imprecision introduced by quantization, compared to datasets with tailed distributions. We leave more in-depth analysis of how graph topology affects quantization as future work.

| Quant. Scheme | Model Arch. | Node Classification (Accuracy %) | | Graph Classification (Accuracy %) | | Graph Regression (Loss) |
|---|---|---|---|---|---|---|
| | | Cora ↑ | Citeseer ↑ | MNIST ↑ | CIFAR-10 ↑ | ZINC ↓ |
| Ref. (FP32) | GCN | $81.4 \pm 0.7$ | $71.1 \pm 0.7$ | $90.0 \pm 0.2$ | $54.5 \pm 0.1$ | $0.469 \pm 0.002$ |
| | GAT | $83.1 \pm 0.4$ | $72.5 \pm 0.7$ | $95.6 \pm 0.1$ | $65.4 \pm 0.4$ | $0.463 \pm 0.002$ |
| | GIN | $77.6 \pm 1.1$ | $66.1 \pm 0.9$ | $93.9 \pm 0.6$ | $53.3 \pm 3.7$ | $0.414 \pm 0.009$ |
| Ours (FP32) | GCN | $81.2 \pm 0.6$ | $71.4 \pm 0.9$ | $90.9 \pm 0.4$ | $58.4 \pm 0.5$ | $0.450 \pm 0.008$ |
| | GAT | $83.2 \pm 0.3$ | $72.4 \pm 0.8$ | $95.8 \pm 0.4$ | $65.1 \pm 0.8$ | $0.455 \pm 0.006$ |
| | GIN | $77.9 \pm 1.1$ | $65.8 \pm 1.5$ | $96.4 \pm 0.4$ | $57.4 \pm 0.7$ | $0.334 \pm 0.024$ |
| QAT (W8A8) | GCN | $81.0 \pm 0.7$ | $71.3 \pm 1.0$ | $90.9 \pm 0.2$ | $56.4 \pm 0.5$ | $0.481 \pm 0.029$ |
| | GAT | $81.9 \pm 0.7$ | $71.2 \pm 1.0$ | $95.8 \pm 0.3$ | $66.3 \pm 0.4$ | $0.460 \pm 0.005$ |
| | GIN | $75.6 \pm 1.2$ | $63.0 \pm 2.6$ | $96.7 \pm 0.2$ | $52.4 \pm 1.2$ | $0.386 \pm 0.025$ |
| nQAT (W8A8) | GCN | $81.0 \pm 0.8$ | $70.7 \pm 0.8$ | $91.1 \pm 0.1$ | $56.2 \pm 0.5$ | $0.472 \pm 0.015$ |
| | GAT | $82.5 \pm 0.5$ | $71.2 \pm 0.7$ | $96.0 \pm 0.1$ | $66.7 \pm 0.2$ | $0.459 \pm 0.007$ |
| | GIN | $77.4 \pm 1.3$ | $65.1 \pm 1.4$ | $96.4 \pm 0.3$ | $52.7 \pm 1.4$ | $0.405 \pm 0.016$ |
| DQ (W8A8) | GCN | $81.7 \pm 0.7$ (**+0.7**) | $71.0 \pm 0.9$ (**-0.3**) | $90.9 \pm 0.2$ (**-0.2**) | $56.3 \pm 0.1$ (**-0.1**) | $0.434 \pm 0.009$ (**+9.8**) |
| | GAT | $82.7 \pm 0.7$ (**+0.2**) | $71.6 \pm 1.0$ (**+0.4**) | $95.8 \pm 0.4$ (**-0.2**) | $67.7 \pm 0.5$ (**+1.0**) | $0.456 \pm 0.005$ (**+0.9**) |
| | GIN | $78.7 \pm 1.4$ (**+1.3**) | $67.5 \pm 1.4$ (**+2.4**) | $96.6 \pm 0.1$ (**-0.1**) | $55.5 \pm 0.6$ (**+2.8**) | $0.357 \pm 0.014$ (**+7.5**) |
| QAT (W4A4) | GCN | $77.2 \pm 2.5$ | $64.1 \pm 4.1$ | $70.6 \pm 2.4$ | $38.1 \pm 1.6$ | $0.692 \pm 0.013$ |
| | GAT | $55.6 \pm 5.4$ | $65.3 \pm 1.9$ | $76.3 \pm 1.2$ | $41.0 \pm 1.1$ | $0.655 \pm 0.032$ |
| | GIN | $42.5 \pm 4.5$ | $18.6 \pm 2.9$ | $91.0 \pm 0.6$ | $45.6 \pm 3.6$ | $0.572 \pm 0.02$ |
| nQAT (W4A4) | GCN | $78.1 \pm 1.5$ | $65.8 \pm 2.6$ | $70.9 \pm 1.5$ | $40.1 \pm 0.7$ | $0.669 \pm 0.128$ |
| | GAT | $54.9 \pm 5.6$ | $65.5 \pm 1.7$ | $78.4 \pm 1.5$ | $41.0 \pm 0.6$ | $0.637 \pm 0.012$ |
| | GIN | $45.0 \pm 5.0$ | $34.6 \pm 3.8$ | $91.3 \pm 0.5$ | $48.7 \pm 1.7$ | $0.561 \pm 0.068$ |
| DQ (W4A4) | GCN | $78.3 \pm 1.7$ (**+0.2**) | $66.9 \pm 2.4$ (**+1.1**) | $84.4 \pm 1.3$ (**+13.5**) | $51.1 \pm 0.7$ (**+11.0**) | $0.536 \pm 0.011$ (**+26.2**) |
| | GAT | $71.2 \pm 2.9$ (**+16.3**) | $67.6 \pm 1.5$ (**+2.1**) | $93.1 \pm 0.3$ (**+14.7**) | $56.5 \pm 0.6$ (**+15.5**) | $0.520 \pm 0.021$ (**+20.6**) |
| | GIN | $69.9 \pm 3.4$ (**+24.9**) | $60.8 \pm 2.1$ (**+26.2**) | $95.5 \pm 0.4$ (**+4.2**) | $50.7 \pm 1.6$ (**+2.0**) | $0.431 \pm 0.012$ (**+23.2**) |

**Table 2:** This table is divided into three sets of rows with FP32 baselines at the top. We provide two baselines for INT8 and INT4: standard QAT and stochastic QAT (nQAT). Both are informed by the analysis in 4.1, with nQAT achieving better performance in some cases. Models trained with Degree-Quant (DQ) are always comparable to baselines, and usually substantially better, especially for INT4. **DQ is a stable method which requires little tuning to obtain excellent results across a variety of architectures and datasets.**

## 4.2 Obtaining Quantization baselines

Our FP32 results, which we obtain after extensive hyperparameter tuning, and those from the baselines are shown at the top of table 2. We observed large gains on MNIST, CIFAR10 and, ZINC.

For our QAT-INT8 and QAT-INT4 baselines, we use the quantization configurations informed by our analysis in section 4.1. For Citeseer we use the best resulting setup analysed for Cora, and for CIFAR-10 that from MNIST. Then, the hyperparameters for each experiment were fine tuned individually, including noise rate $n \in [0.5, 0.95]$ for nQAT experiments. QAT-INT8 and QAT-INT4 results in table 2 and QAT-INT4, with the exception of MNIST (an easy to classify dataset), corroborate our hypothesis that GIN layers are less resilient to quantization. This was first observed in fig. 3. In the case of ZINC, while all models results in noticeable degradation, GIN sees a more severe $16\%$ increase of regression loss compared to our FP32 baseline. For QAT W4A4 an accuracy drop of over $35\%$ and $47\%$ is observed for Cora and Citeseer respectively. The stochasticity induced by nQAT helped in recovering some of the accuracy lost as a result of quantization for citation networks (both INT8 and INT4) but had little impact on other datasets and harmed performance in some cases.

## 4.3 Comparisons of Degree-Quant with Existing Quantization Approaches

Degree-Quant provides superior quantization for all GNN datasets and architectures. Our results with DQ are highlighted in gray in table 2 and table 3. Citation networks trained with DQ for W8A8 manage to recover most of the accuracy lost as a result of QAT and outperform most of nQAT baselines. In some instances DQ-W8A8 models outperform the reference FP32 baselines. At 4-bits, DQ results in even larger gains compared to W4A4 baselines. We see DQ being more effective for GIN layers, outperforming INT4 baselines for Cora ($+24.9\%$), Citeseer ($+26.2\%$) and REDDIT-BINARY ($+23.0\%$) by large margins. Models trained with DQ at W4A4 for graph classification and graph regression also exhibit large performance gains (of over $10\%$) in most cases. For ZINC, all

| Quantization | Model | REDDIT-BIN (Acc. %) ↑ |
|---|---|---|
| Ref. (FP32) | GIN | $92.2 \pm 2.3$ |
| Ours (FP32) | GIN | $92.0 \pm 1.5$ |
| QAT-W8A8 | GIN | $76.1 \pm 7.5$ |
| nQAT-W8A8 | GIN | $77.5 \pm 3.4$ |
| DQ-W8A8 | GIN | $91.8 \pm 2.3$ (+**14.3**) |
| QAT-W4A4 | GIN | $54.4 \pm 6.6$ |
| nQAT-W4A4 | GIN | $58.0 \pm 6.3$ |
| DQ-W4A4 | GIN | $81.3 \pm 4.4$ (+**23.0**) |

**Table 3:** Results for DQ-INT8 GIN models perform nearly as well as at FP32. For INT4, DQ offers a significant increase in accuracy.

| Device | Arch. | Zinc (Batch=10K) | | | Reddit | | |
|---|---|---|---|---|---|---|---|
| | | FP32 | W8A8 | Speedup | FP32 | W8A8 | Speedup |
| CPU | GCN | 181ms | 42ms | 4.3× | 13.1s | 3.1s | 4.2× |
| | GAT | 190ms | 50ms | 3.8× | 13.1s | 2.8s | 4.7× |
| | GIN | 182ms | 43ms | 4.2× | 13.1s | 3.1s | 4.2× |
| GPU | GCN | 39ms | 31ms | 1.3× | 191ms | 176ms | 1.1× |
| | GAT | 17ms | 15ms | 1.1× | OOM | OOM | - |
| | GIN | 39ms | 31ms | 1.3× | 191ms | 176ms | 1.1× |

**Table 4:** INT8 latency results run on a 22 core 2.1GHz Intel Xeon Gold 6152 and, on a GTX 1080Ti GPU. Quantization provides large speedups on a variety of graphs for CPU and non-negligible speedups with unoptimized INT8 GPU kernels.

models achieve over $20\%$ lower regression loss. Among the top performing models using DQ, ratios of $p_{min}$ and $p_{max}$ in $[0.0, 0.2]$ were the most common. Figure 10 in the appendix shows validation loss curves for GIN models trained using different DQ probabilities on the REDDIT-BINARY dataset.

## 5 DISCUSSION

**Latency and Memory Implications**. In addition to offering significantly lower memory usage ($4\times$ with INT8), quantization can reduce latency—especially on CPUs. We found that with INT8 arithmetic we could accelerate inference by up to $4.7\times$. We note that the latency benefit depends on the graph topology and feature dimension, therefore we ran benchmarks on a variety of graph datasets, including Reddit[2], Zinc, Cora, Citeseer, and CIFAR-10; Zinc and Reddit results are shown in table 4, with further results given in the appendix. For a GCN layer with in- and out-dimension of 128, we get speed-ups of: $4.3\times$ on Reddit, $2.5\times$ on Zinc, $1.3\times$ on Cora, $1.3\times$ on Citeseer and, $2.1\times$ on CIFAR-10. It is also worth emphasizing that quantized networks are necessary to efficiently use accelerators deployed in smartphones and smaller devices as they primarily accelerate integer arithmetic, and that CPUs remain a common choice for model serving on servers. The decrease in latency on CPUs is due to improved cache performance for the sparse operations; GPUs, however, see less benefit due to their massively-parallel nature which relies on mechanisms other than caching to hide slow random memory accesses, which are unavoidable in this application.

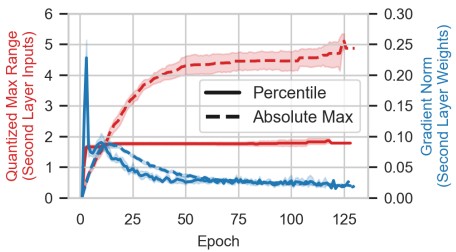
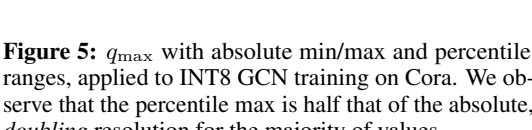
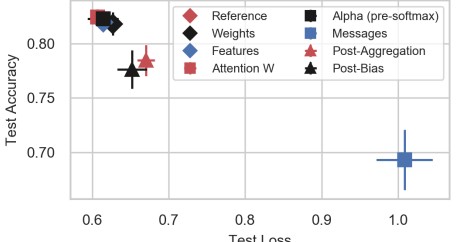

**Figure 5:** $q_{max}$ with absolute min/max and percentile ranges, applied to INT8 GCN training on Cora. We observe that the percentile max is half that of the absolute, *doubling* resolution for the majority of values.

**Figure 6:** Analysis of how INT8 GAT performance degrades on Cora as individual elements are reduced to 4-bit precision *without DQ*. For GAT the message elements are crucial to classification performance.

**Ablation Study: Benefits of Percentile Ranges**. Figure 5 shows the value of percentiles during training. We see that when using absolute min/max the upper range grows to over double the range required for 99.9% of values, effectively halving the resolution of the quantized values. DQ is more stable, and we obtained strong results with an order of magnitude less tuning relative to the baselines.

**Ablation Study: Source of Degradation at INT4**. Figure 6 assesses how INT8 GAT (without DQ) degrades as single elements are converted to INT4, in order to understand the precipitous drop in

[2]The largest graph commonly benchmarked on in the GNN literature

accuracy in the INT4 baselines; further plots for GCN and GIN are included in the appendix. We observe that most elements cause only modest performance losses relative to a full INT8 model. DQ is most important to apply to elements which are constrained by *numerical precision*, such as the aggregation and message elements in GAT. Weight elements, however, are consistently unaffected.

**Ablation Study: Effect of Stochastic Element in Degree-Quant**. We observe that the stochastic protective masking in DQ alone often achieves most of the performance gain over the QAT baseline; results are given in table 9 in the appendix. The benefit of the percentile-based quantization ranges is *stability*, although it can yield some performance gains. The full DQ method provides consistently good results on all architectures and datasets, without requiring an extensive analysis as in 4.1.

## 6 CONCLUSION

This work has presented Degree-Quant, an architecture-agnostic and stable method for training quantized GNN models that can be accelerated using off-the-shelf hardware. With 4-bit weights and activations we achieve $8\times$ compression while surpassing strong baselines by margins regularly exceeding 20%. At 8-bits, models trained with DQ perform on par or better than the baselines while achieving up to $4.7\times$ lower latency than FP32 models. Our work offers a comprehensive foundation for future work in this area and is a first step towards enabling GNNs to be deployed more widely, including to resource constrained devices such as smartphones.

## ACKNOWLEDGEMENTS

This work was supported by Samsung AI and by the UK's Engineering and Physical Sciences Research Council (EPSRC) with grants EP/M50659X/1 and EP/S001530/1 (the MOA project) and the European Research Council via the REDIAL project.

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

# A    APPENDIX

Readers seeking advice on implementation will find appendix A.5 especially useful. We provide significant advice surrounding best practices on quantization for GNNs, along with techniques which we believe can boost our methods beyond the performance described in this paper, but for which we did not have time to fully evaluate.

## A.1    EXPERIMENTAL SETUP

As baselines we use the architectures and results reported by Fey & Lenssen (2019) for citation networks, Dwivedi et al. (2020) for MNIST, CIFAR-10 and ZINC and, Xu et al. (2019) for REDDIT-BINARY. We re-implemented the architectures and datasets used in these publications and replicated the results reported at FP32. Models using GIN layers learn parameter $\epsilon$. These models are often referred to as GIN-$\epsilon$. The high-level description of these architectures is shown in table 5. The number of parameters for each architecture-dataset in this work are shown in table 6.

Our infrastructure was implemented using PyTorch Geometric (PyG) (Fey & Lenssen, 2019). We generate candidate hyperparameters using random search, and prune trials using the asynchronous hyperband algorithm (Li et al., 2020). Hyperparameters searched over were learning rate, weight decay, and dropout (Srivastava et al., 2014) and drop-edge (Rong et al., 2020) probabilities. The search ranges were initialized centered at the values used in the reference implementations of the baselines. Degree-Quant requires searching for two additional hyperparameters, $p_{min}$ and $p_{max}$, these were tuned in a grid-search fashion. We report our results using the hyperparameters which achieved the best validation loss over 100 runs on the Cora and Citeseer datasets, 10 runs for MNIST, CIFAR-10 and ZINC, and 10-fold cross-validation for REDDIT-BINARY.

We generally used fewer hyperparameter runs for our DQ runs than we did for baselines—even ignoring the searches over the various STE configs. As our method is more stable, finding a reasonable set of parameters was easier than before. As is usual with quantization experiments, we found it useful to decrease the learning rate relative to the FP32 baseline.

Our experiments ran on several machines in our SLURM cluster using Intel CPUs and NVIDIA GPUs. Each machine was running Ubuntu 18.04. The GPU models in our cluster were: V100, RTX 2080Ti and GTX 1080Ti.

| Model | # Layers | | | | | # Hidden Units | | | | | Residual | | | | | Output MLP | | | | |
|---|---|---|---|---|---|---|---|---|---|---|---|---|---|---|---|---|---|---|---|---|
| Arch. | Cit | M | C | Z | R | Cit | M | C | Z | R | Cit | M | C | Z | R | Cit | M | C | Z | R |
| GCN | 2 | 4 | 4 | 4 | - | 16 | 146 | 146 | 145 | - | × | ✓ | ✓ | ✓ | - | × | ✓ | ✓ | ✓ | - |
| GAT | 2 | 4 | 4 | 4 | - | 8 | 19 | 19 | 18 | - | × | ✓ | ✓ | ✓ | - | × | ✓ | ✓ | ✓ | - |
| GIN | 2 | 4 | 4 | 4 | 5 | 16 | 110 | 110 | 110 | 64 | × | ✓ | ✓ | ✓ | × | × | ✓ | ✓ | ✓ | ✓ |

**Table 5:** High level description of the architectures evaluated for citation networks (Cit), MNIST (M), CIFAR-10 (C), ZINC (Z) and REDDIT-BINARY (R). We relied on Adam optimizer for all experiments. For all batched experiments, we used 128 batch-sizes. All GAT models used 8 attention heads. All GIN architectures used 2-layer MLPs, except those for citation networks which used a single linear layer.

| Model | Node Classification | | Graph Classification | | | Graph Regression |
|---|---|---|---|---|---|---|
| Arch. | Cora | Citeseer | MNIST | CIFAR-10 | REDDIT-BIN | ZINC |
| GCN | 23063 | 59366 | 103889 | 104181 | - | 105454 |
| GAT | 92373 | 237586 | 113706 | 114010 | - | 105044 |
| GIN | 23216 | 59536 | 104554 | 104774 | 42503 | 102088 |

**Table 6:** Number of parameters for each of the evaluated architectures

For QAT experiments, all elements of each network are quantized: inputs to each layer, the weights, the messages sent between nodes, the inputs to aggregation stage and its outputs and, the outputs of the update stage (which are the outputs of the GNN layer before activation). In this way, all intermediate tensors in GNNs are quantized with the exception of the attention mechanism in GAT; we do not quantize after the softmax calculation, due to the numerical precision required at this

stage. With the exception of Cora and Citeseer, the models evaluated in this work make use of Batch Normalization (Ioffe & Szegedy, 2015). For deployments of quantized models, Batch Normalization layers are often *folded* with the weights (Krishnamoorthi, 2018). This is to ensure the input to the next layer is within the expected $[q_{min}, q_{max}]$ ranges. In this work, for both QAT baselines and QAT+DQ, we left BN layers unfolded but ensure the inputs and outputs were quantized to the appropriate number of bits (i.e. INT8 or INT4) before getting multiplied with the layer weights. We leave as future work proposing a BN folding mechanism applicable for GNNs and studying its impact for deployments of quantized GNNs.

The GIN models evaluated on REDDIT-BINARY used QAT for all layers with the exception of the input layer of the MLP in the first GIN layer. This compromise was needed to overcome the severe degradation introduced by quantization when operating on nodes with a single scalar as feature.

## A.2 DATASETS

We show in Table 7 the statistics for each dataset either used or referred to in this work. For Cora and Citeseer datasets, nodes correspond to documents and edges to citations between these. Node features are a bag-of-words representation of the document. The task is to classify each node in the graph (i.e. each document) correctly. The MNIST and CIFAR-10 datasets (commonly used for image classification) are transformed using SLIC (Achanta et al., 2012) into graphs where each node represents a cluster of perceptually similar pixels or superpixels. The task is to classify each image using their superpixels graph representation. The ZINC dataset contains graphs representing molecules, were each node is an atom. The task is to regress a molecular property (constrained solubility (Jin et al., 2018)) given the graph representation of the molecule. Nodes in graphs of the REDDIT-BINARY dataset represent users of a Reddit thread with edges drawn between a pair of nodes if these interacted. This dataset contains graphs of two types of communities: question-answer threads and discussion threads. The task is to determine if a given graph is from a question-answer thread or a discussion thread.

We use standard splits for MNIST, CIFAR-10 and ZINC. For citation datasets (Cora and Citeseer), we use the splits used by Kipf & Welling (2017). For REDDIT-BINARY we use 10-fold cross validation.

| Dataset | Graphs | Nodes | Edges | Features | Labels |
|---|---|---|---|---|---|
| Cora | 1 | 2,708 | 5,278 | 1,433 | 7 |
| Citeseer | 1 | 3,327 | 4,552 | 3,703 | 6 |
| Pubmed | 1 | 19,717 | 44,324 | 500 | 3 |
| MNIST | 70K | 40-75 | 564.53 (avg) | 3 | 10 |
| CIFAR10 | 60K | 85-150 | 941.07 (avg) | 5 | 10 |
| ZINC | 12K | 9-37 | 49.83 (avg) | 28 | 1 |
| REDDIT-BINARY | 2K | 429.63 (avg) | 497.75 (avg) | 1 | 2 |
| Reddit | 1 | 232,965 | 114,848,857 | 602 | 41 |
| Amazon | 1 | 9,430,088 | 231,594,310 | 300 | 24 |

**Table 7:** Statistics for each dataset used in the paper. Some datasets are only referred to in fig. 1

## A.3 QUANTIZATION IMPLEMENTATIONS

In section 4.1 we analyse different readily available quantization implementations and how they impact in QAT results. First, vanilla STE, which is the reference STE (Bengio et al., 2013) that lets the gradients pass unchanged; and gradient clipping (GC), which clips the gradients based on the maximum representable value for a given quantization level. Or in other words, GC limits gradients if the tensor's magnitudes are outside the $[q_{min}, q_{max}]$ range.

$$x_{min} = \begin{cases} \min(X) & \text{if step} = 0 \\ \min(x_{min}, X) & \text{otherwise} \end{cases} \tag{1}$$

$$x_{min} = \begin{cases} \min(X) & \text{if step} = 0 \\ (1 - c)x_{min} + c\min(X) & \text{otherwise} \end{cases} \tag{2}$$

The quantization modules keep track of the input tensor's min and max values, $x_{\min}$ and $x_{\max}$, which are then used to compute $q_{\min}$, $q_{\max}$, *zero-point* and *scale* parameters. For both vanilla STE and GC, we study two popular ways of keeping track of these statistics: *min/max*, which tracks the min/max tensor values observed over the course of training; and *momentum*, which computes the moving averages of those statistic during training. The update rules for $x_{\min}$ for STE *min/max* and STE *momentum* are presented in eq. (1) and eq. (2) respectively, where $X$ is the tensor to be quantized and $c$ is the momentum hyperparameter, which in all our experiments is set to its default 0.01. Equivalent rules apply when updating $x_{\max}$ (omitted).

For stochastic QAT we followed the implementation described in Fan et al. (2020), where at each training step a binary mask sampled from a Bernoulli distribution is used to specify which elements of the weight tensor will be quantized and which will be left at full precision. We experimented with block sizes larger than one (i.e. a single scalar) but often resulted in a sever drop in performance. All the reported results use block size of one.

## A.4   Degree-Quant and Graph Level Summarization

The percentile operation in our quantization scheme remains important for summarizing the graph when doing graph-level tasks, such as graph regression (Zinc) or graph classification (MNIST, CIFAR-10 and REDDIT-BINARY). Since the number of nodes in each input graph is not constant, this can cause the summarized representation produced from the final graph layer to have a more tailed distribution than would be seen with other types of architectures (e.g. CNN). Adding the percentile operation reduces the impact of these extreme tails in the fully connected graph-summarization layers, thereby increasing overall performance. The arguments regarding weight update accuracy also still apply, as the $\frac{\partial \mathcal{L}}{\partial \mathbf{h}_{l+1}^{(i)}}$ term in the equations for the GCN and GIN should be more accurate compared to when the activations are always quantized before the summarization. This phenomenon is also noted by Fan et al. (2020).

## A.5   Implementation Advice

We provide details that will be useful for others working in the area, including suggestions that should boost the performance of our results and accelerate training. We release code on GitHub; this code is a clean implementation of the paper, suitable for users in downstream works.

### A.5.1   Quantization Setup

As our work studies the pitfalls of quantization for GNNs, we were more aggressive in our implementation than is absolutely necessary: everything (where reasonably possible) in our networks is quantized. In practice, this leaves low-hanging fruit for improvements in accuracy:

- Not quantizing the final layer (as is common practice for CNNs and Transformers) helps with accuracy, especially at INT4. A similar practice at the first layer will also be useful.

- Using higher precision for the "summarization" stages of the model, which contributes little towards the runtime in most cases.

- Taking advantage of mixed precision: since the benefits of quantization are primarily in the message passing phase (discussed below), one technique to boost accuracy is to only make the messages low precision.

We advise choosing a more realistic (less aggressive) convention than used in this work. The first two items would be appropriate.

### A.5.2   Relative Value of Percentiles Compared to Protective Masking

There are two components to our proposed technique: stochastic, topology-aware, masking and percentile-based range observers for quantizers. We believe that percentiles provide more immediate value, especially at INT4. We find that they are useful purely from the perspective of stabilizing the optimization and reducing the sensitivity to hyperparameters.

However, adding the masking does improve performance further. This is evident from table 9. In fact, performance may be degraded slightly when percentiles are also applied: this can be observed by comparing table 9 to the main results in the paper, table 2.

### A.5.3 PERCENTILES

The key downside with applying percentiles for range observers is that the operation can take significant time. Training with DQ is slower than before—however, since there is less sensitivity to hyperparameters, fewer runs end up being needed. We are confident that an effective way to speed up this operation is to use sampling. We expect 10% of the data should be adequate, however we believe that even 1% of the data may be sufficient (dataset and model dependent). However, we have not evaluated this setup in the paper; it is provided in the code release for experimentation.

### A.5.4 IMPROVING ON PERCENTILES

We believe that it is possible to significantly boost the performance of GNN quantization by employing a learned step size approach. Although we used percentiles in this paper to illustrate the range-precision trade-off for GNNs, we expect that *learning* the ranges will lead to better results. This approach, pioneered by works such as Esser et al. (2020), has been highly effective on CNNs even down to 2 bit quantization.

Another approach would be to use *robust quantization*: the ideas in these works are to reduce the impact of changing quantization ranges i.e. making the architecture more robust to quantization. Works in this area include Alizadeh et al. (2020) and Shkolnik et al. (2020).

### A.5.5 IMPROVING LATENCY

The slowest step of GNN inference is typically the sparse operations. It is therefore best to minimize the sizes of the messages between nodes i.e. quantize the message phase most aggressively. This makes the biggest impact on CPUs which are dependent on caches to obtain good performance.

We evaluated our code on CPU using Numpy and Scipy routines. For the GPU, we used implementations from PyTorch and PyTorch Geometric and lightly modified them to support INT8 where necessary. These results, while useful for illustrating the benefits of quantization, are by no means optimal: we did not devote significant time to improving latency. We believe better results can be obtained by taking advantage of techniques such as cache blocking or kernel fusion.

### A.5.6 PITFALLS

Training these models can be highly unstable: some experiments in the paper had standard deviations as large as 18%. We observed this to affect citation network experiments to the extent that they would not converge on GPUs: all these experiments had to be run on CPUs.

### A.6 DEGRADATION STUDIES

Figures 7 and 8 show the results of the ablation study conducted in section 5 for GCN and GIN. We observe that GCN is more tolerant to INT4 quantization than other architectures. GIN, however, requires accurate representations after the update stage, and heavily suffers from further quantization like GAT. The idea of performing different stages of inference at different precisions has been proposed, although it is uncommon (Wang et al., 2018).

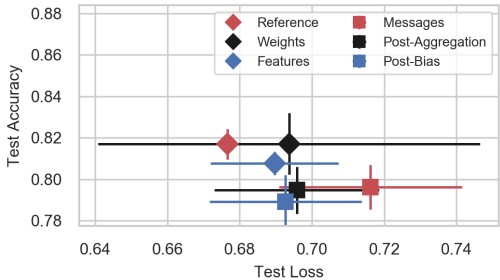

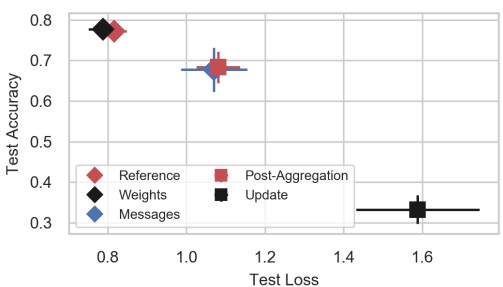

**Figure 7:** Degradation of INT8 GCN on Cora as individual elements are converted to INT4 *without Degree-Quant*.

**Figure 8:** Degradation of INT8 GIN on Cora as individual elements are converted to INT4 *without Degree-Quant*.

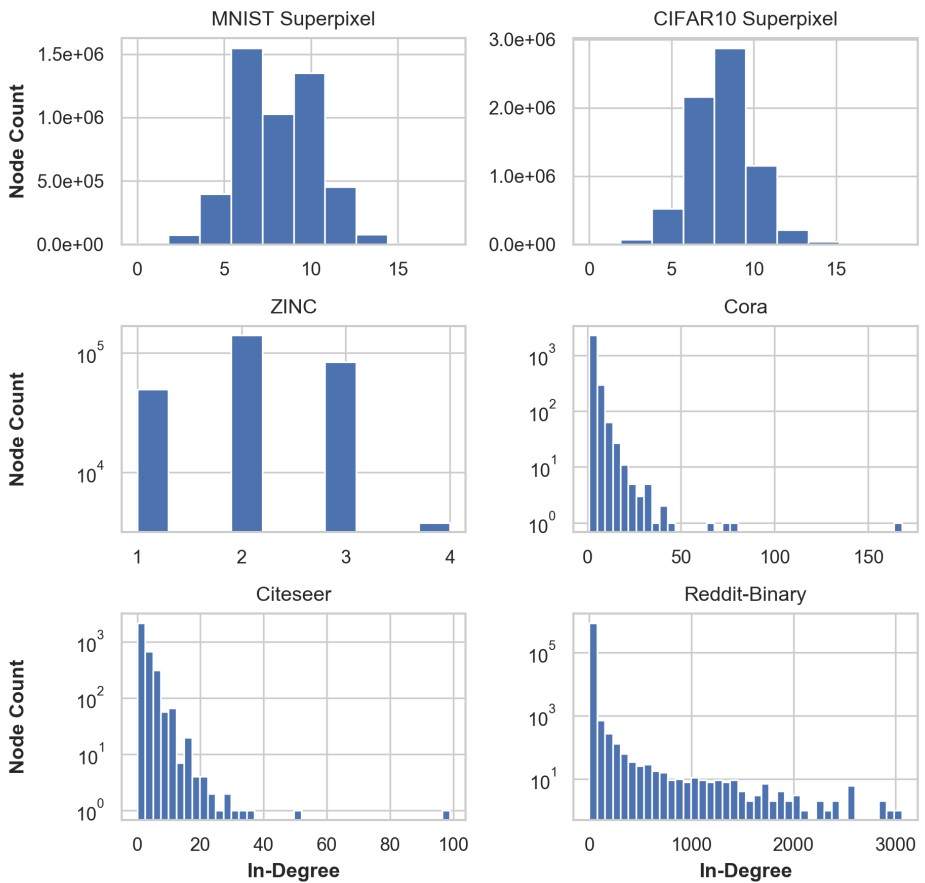

**Figure 9:** In-degree distribution for each of the six datasets assessed. Note that a log $y$-axis is used for all datasets except for MNIST and CIFAR-10.

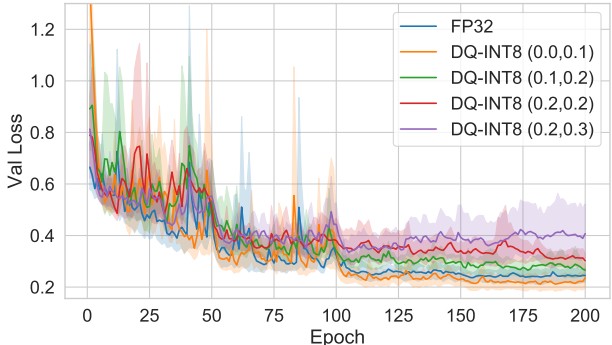

| Quantization | Model | REDDIT-BIN ↑ |
|---|---|---|
| Ref. (FP32) | GIN | $92.2 \pm 2.3$ |
| Ours (FP32) | GIN | $92.0 \pm 1.5$ |
| DQ-INT8 (0.0, 0.1) | GIN | $91.8 \pm 2.3$ |
| DQ-INT8 (0.1, 0.2) | GIN | $90.1 \pm 2.5$ |
| DQ-INT8 (0.2, 0.2) | GIN | $89.0 \pm 3.0$ |
| DQ-INT8 (0.2, 0.3) | GIN | $88.1 \pm 3.0$ |

**Figure 10:** Validation loss curves for GIN models evaluated on REDDIT-BINARY. Results averaged across 10-fold cross-validation. We show four DQ-INT8 experiments each with a different values for $(p_{\min}, p_{\max})$ and our FP32 baseline.

**Table 8:** Final test accuracies for FP32 and DQ-INT8 models whose validation loss curves are shown in fig. 10

| Quantization Scheme | Model Arch. | Node Classification Cora ↑ | Node Classification Citeseer ↑ | Graph Regression ZINC ↓ |
|---|---|---|---|---|
| | GCN | $81.1 \pm 0.6$ | $71.0 \pm 0.7$ | $0.468 \pm 0.014$ |
| QAT-INT8 + DQ Masking | GAT | $82.1 \pm 0.1$ | $71.4 \pm 0.8$ | $0.462 \pm 0.005$ |
| | GIN | $78.9 \pm 1.2$ | $67.1 \pm 1.7$ | $0.347 \pm 0.028$ |
| | GCN | $78.5 \pm 1.4$ | $62.8 \pm 8.5$ | $0.599 \pm 0.015$ |
| QAT-INT4 + DQ Masking | GAT | $64.4 \pm 9.3$ | $68.9 \pm 1.2$ | $0.529 \pm 0.008$ |
| | GIN | $71.2 \pm 2.9$ | $56.7 \pm 3.8$ | $0.427 \pm 0.010$ |
| | GCN | $75.6 \pm 2.5$ | $64.8 \pm 3.8$ | $0.633 \pm 0.012$ |
| nQAT-INT4 + Percentile | GAT | $70.1 \pm 2.8$ | $51.4 \pm 3.4$ | $0.596 \pm 0.008$ |
| | GIN | $63.5 \pm 2.0$ | $46.3 \pm 4.1$ | $0.771 \pm 0.058$ |

**Table 9:** Ablation study against the two elements of Degree-Quant (DQ). The first two rows of results are obtained with only the stochastic element of Degree-Quant enabled for INT8 and INT4. Percentile-based quantization ranges are disabled in these experiments. The bottom row of results were obtained with noisy quantization (nQAT) at INT4 with the use of percentiles. DQ masking alone is often sufficient to achieve excellent results, but the addition of percentile-based range tracking can be beneficial to increase stability. We can see that using nQAT with percentiles is not sufficient to achieve results of the quality DQ provides.

| Device | Arch. | CIFAR-10 FP32 | CIFAR-10 W8A8 | CIFAR-10 Speedup | Cora FP32 | Cora W8A8 | Cora Speedup | Citeseer FP32 | Citeseer W8A8 | Citeseer Speedup |
|---|---|---|---|---|---|---|---|---|---|---|
| | GCN | 182ms | 88ms | 2.1× | 0.94ms | 0.74ms | 1.3× | 0.97ms | 0.76ms | 1.3× |
| CPU | GAT | 500ms | 496ms | 1.0× | 0.86ms | 0.78ms | 1.1× | 0.99ms | 0.88ms | 1.1× |
| | GIN | 144ms | 44ms | 3.3× | 0.85ms | 0.68ms | 1.3× | 0.95ms | 0.55ms | 1.7× |
| | GCN | 2.1ms | 1.6ms | 1.3× | 0.08ms | 0.09ms | 0.9× | 0.09ms | 0.09ms | 1.0× |
| GPU | GAT | 30.0ms | 27.1ms | 1.1× | 0.57ms | 0.64ms | 0.9× | 0.56ms | 0.64ms | 0.9× |
| | GIN | 20.9ms | 16.2ms | 1.2× | 0.09ms | 0.07ms | 1.3× | 0.09ms | 0.07ms | 1.3× |

**Table 10:** INT8 latency results run on a 22 core 2.1GHz Intel Xeon Gold 6152 and, on a GTX 1080Ti GPU. All layers have 128 in/out features. For CIFAR-10 we used batch size of 1K graphs.

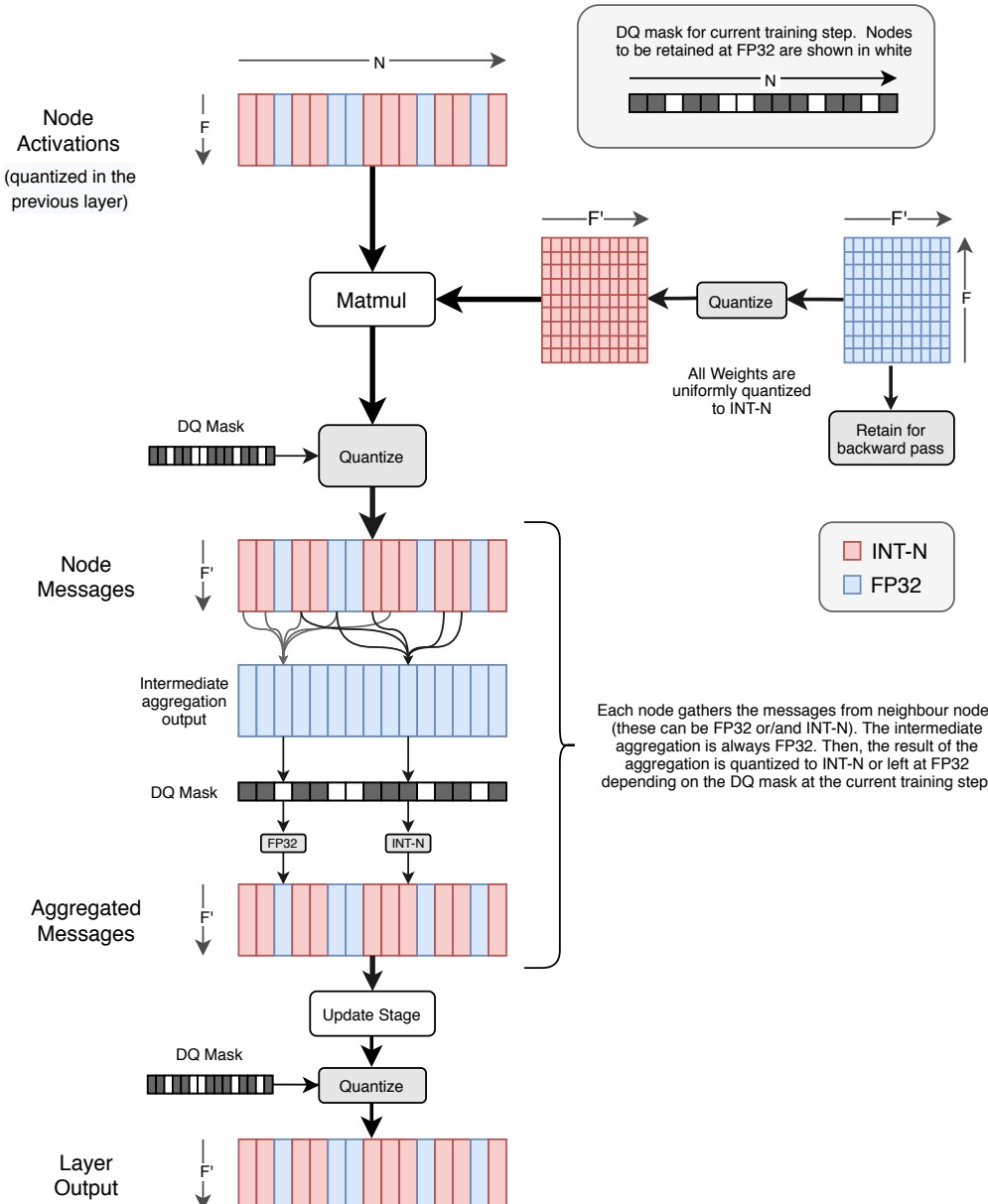

**Figure 11:** Diagram representing how DQ makes use of a topology-aware quantization strategy that is better suited for GNNs. The diagram illustrates this for a GCN layer. At every training stage, a degree-based mask is generated. This mask is used in all quantization layers located after each of the stages in the message-passing pipeline. By retaining at FP32 nodes with higher-degree more often, the noisy updates during training have a lesser impact and therefore models perform better, even at INT4.

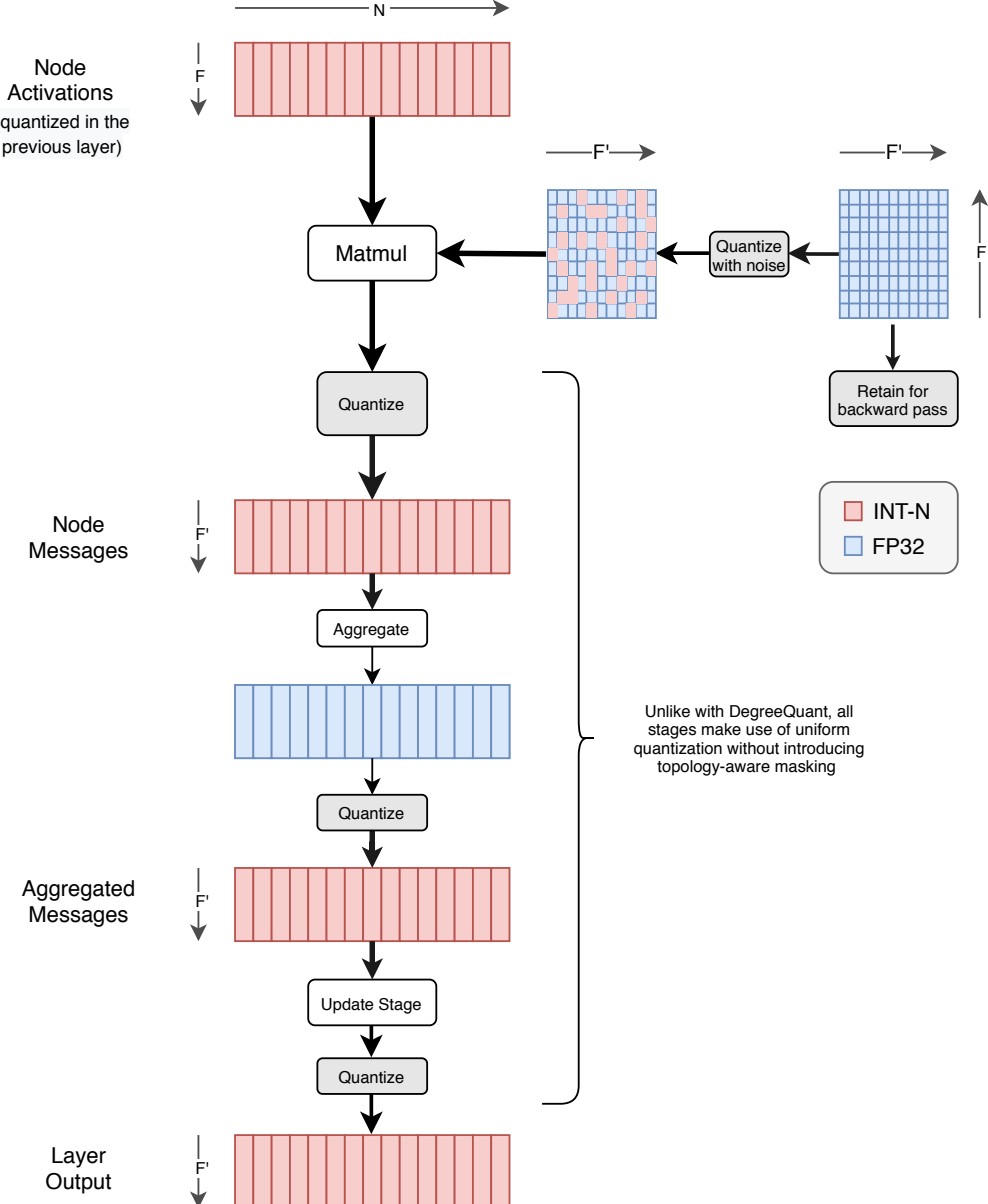

**Figure 12:** Diagram representing how nQAT is implemented for GNNs. The diagram illustrates this for a GCN layer. The stochastic stage only takes place when quantizing the weights, the remaining of the quantization modules happen following a standard QAT strategy. A QAT diagram would be similar to this one but fully quantizing the weights.

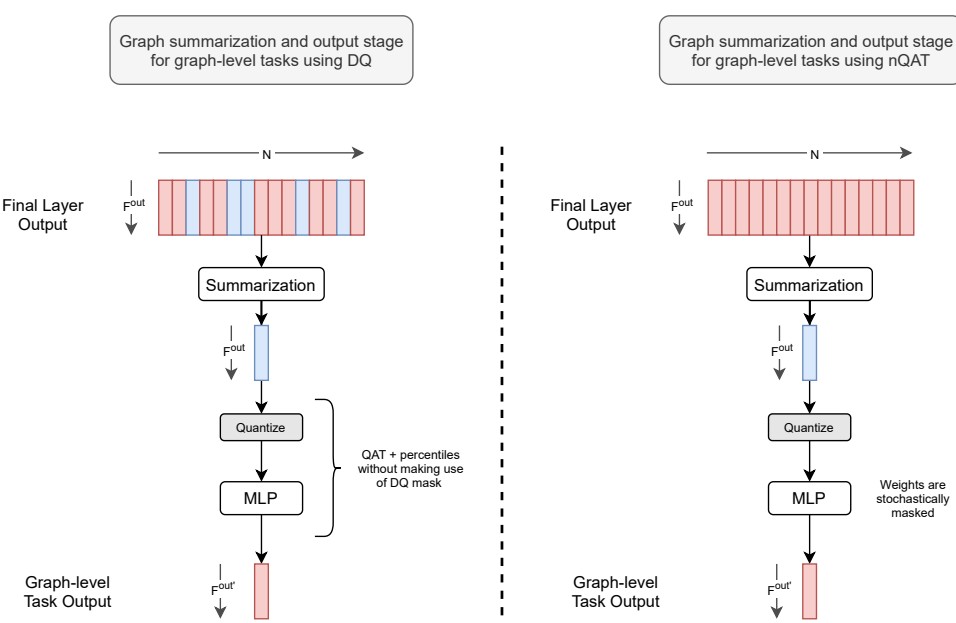

**Figure 13:** Diagrams representing how the output graph-summarization stages for graph-level tasks (e.g. graph classification, graph regression) are implemented when making use of DQ (left) and nQAT (right). GNNs making use of DQ during the node-aggregation stages (see fig. 11), do not use the stochastic element of DQ in the output MLP layers but still make use of percentiles. For models making use of nQAT, the final MLP still makes use of stochastic quantization of weights.

## B  CODE LISTINGS

Our code depends on PyTorch Geometric (Fey & Lenssen, 2019). These snippets should be compatible with Python 3.7 and PyTorch Geometric version 1.4.3. You can see the full code on GitHub: `https://github.com/camlsys/degree-quant`.

### B.1  MASK GENERATION

```python
class ProbabilisticHighDegreeMask:
    def __init__(self, low_prob, high_prob, per_graph=True):
        self.low_prob = low_prob
        self.high_prob = high_prob
        self.per_graph = per_graph

    def _process_graph(self, graph):
        # Note that:
        # 1. The probability of being masked increases as the indegree increases
        # 2. All nodes with the same indegree have the same bernoulli p
        # 3. you can set this such that all nodes have some probability of being masked

        n = graph.num_nodes
        indegree = degree(graph.edge_index[1], n, dtype=torch.long)
        counts = torch.bincount(indegree)

        step_size = (self.high_prob - self.low_prob) / n
        indegree_ps = counts * step_size
        indegree_ps = torch.cumsum(indegree_ps, dim=0)
        indegree_ps += self.low_prob
        graph.prob_mask = indegree_ps[indegree]

        return graph

    def __call__(self, data):
        if self.per_graph and isinstance(data, Batch):
            graphs = data.to_data_list()
            processed = []
            for g in graphs:
                g = self._process_graph(g)
                processed.append(g)
            return Batch.from_data_list(processed)
        else:
            return self._process_graph(data)

def evaluate_prob_mask(data):
    return torch.bernoulli(data.prob_mask).to(torch.bool)
```

### B.2  MESSAGE PASSING WITH DEGREE-QUANT

Here we provide code to implement the layers as used by our proposal. These are heavily based off of the classes provided by PyTorch Geometric, with only minor modifications to insert the quantization steps where necessary. The normal quantized versions are similar, except without any concept of high/low masking.

```python
class MessagePassingMultiQuant(nn.Module):
    """This class is a lightweight modification of the default PyTorch
    Geometric MessagePassing class"""

    # irrelevant methods removed

    def propagate(self, edge_index, mask, size=None, **kwargs):
        # some lines skipped ...
        msg = self.message(**msg_kwargs)
        if self.training:
            # This is for the masking of messages:
            edge_mask = torch.index_select(mask, 0, edge_index[0])
            out = torch.empty_like(msg)
            out[edge_mask] = self.mp_quantizers["message_high"](msg[edge_mask])
            out[~edge_mask] = self.mp_quantizers["message_low"](
                msg[~edge_mask]
            )
        else:
            out = self.mp_quantizers["message_low"](msg)

        aggr_kwargs = self.__distribute__(self.__aggr_params__, kwargs)
```

```python
        aggrs = self.aggregate(out, **aggr_kwargs)
        if self.training:
            out = torch.empty_like(aggrs)
            out[mask] = self.mp_quantizers["aggregate_high"](aggrs[mask])
            out[~mask] = self.mp_quantizers["aggregate_low"](aggrs[~mask])
        else:
            out = self.mp_quantizers["aggregate_low"](aggrs)

        update_kwargs = self.__distribute__(self.__update_params__, kwargs)
        updates = self.update(out, **update_kwargs)
        if self.training:
            out = torch.empty_like(updates)
            out[mask] = self.mp_quantizers["update_high"](updates[mask])
            out[~mask] = self.mp_quantizers["update_low"](updates[~mask])
        else:
            out = self.mp_quantizers["update_low"](updates)

        return out
```

### B.2.1 GCN

```python
class GCNConvMultiQuant(MessagePassingMultiQuant):
    # Some methods missed...
    def forward(self, x, edge_index, mask, edge_weight=None):
        # quantizing input
        if self.training:
            x_q = torch.empty_like(x)
            x_q[mask] = self.layer_quantizers["inputs_high"](x[mask])
            x_q[~mask] = self.layer_quantizers["inputs_low"](x[~mask])
        else:
            x_q = self.layer_quantizers["inputs_low"](x)

        # quantizing layer weights
        w_q = self.layer_quantizers["weights_low"](self.weight)
        if self.training:
            x = torch.empty((x_q.shape[0], w_q.shape[1])).to(x_q.device)
            x_tmp = torch.matmul(x_q, w_q)
            x[mask] = self.layer_quantizers["features_high"](x_tmp[mask])
            x[~mask] = self.layer_quantizers["features_low"](x_tmp[~mask])
        else:
            x = self.layer_quantizers["features_low"](torch.matmul(x_q, w_q))

        if self.normalize:
            edge_index, norm = self.norm(
                edge_index,
                x.size(self.node_dim),
                edge_weight,
                self.improved,
                x.dtype,
            )
        else:
            norm = edge_weight

        norm = self.layer_quantizers["norm"](norm)
        return self.propagate(edge_index, x=x, norm=norm, mask=mask)
```

