# OpenReview forum: "Degree-Quant: Quantization-Aware Training for Graph Neural Networks"
_ICLR.cc/2021/Conference — ICLR 2021 Poster_

### Official Review · AnonReviewer4 · 2020-10-28
**Good paper; clarity**

**Rating:** 6
**Confidence:** 2

**Review:**

(Edit: Sorry, the previous review was for a different paper that ended up in here due to a copy-paste issue)

This paper uses quantization and Quantization Aware Training (QAT) to improve the speed performance of GNN inference for three types of GNN modes: GIN, GCN and GAT. The paper identifies the aggregation step to be where quantization introduces the most numerical error, and use stochastic masking and clipping the top/bottom values to mitigate the issue.

This topic is very relevant and interesting, and novel to the best of my knowledge---although I'm not familiar with the literature surrounding quantized neural networks.

There are places where the writing can be more careful. For example, in the abstract the authors write: "little research exploring methods to make GNN more efficient at inference time". However, there has been research focusing on both hardware acceleration [1] and making GNN models smaller [2]. Quantization isn't the only approach to make GNN inference faster. Claims like "it is not possible to deploy this technique on smartphones" (from intro paragraph 2) should be supported, since it's difficult for a reader
to verify such a claim.

Some of the claims, like the one bolded in Table 1, should be in the abstract. I'm not sure if this is typical in the quantization literature, but a wallclock time comparison would be useful in Table 2 to compare the time speedup against the baseline.

One other presentation feedback: in figure 1, the x-axis is not continuous. A line chart is not appropriate since the slope of the line segments in the chart is meaningless. Removing the lines connecting the dots would make more sense.

[1] Zeng and Prasanna. (2020) GraphACT: Accelerating GCN training on CPU-FPGA heterogeneous platform https://arxiv.org/abs/2001.02498
[2] Yan et al. (2020) TinyGNN: Learning Efficient Graph Neural Networks https://dl.acm.org/doi/abs/10.1145/3394486.3403236

---

> ### Author Response · Authors · 2020-11-16
> **improved quality of writing; addressing of concerns and discussion of quantization benefits**
>
> Thank you for taking the time to review our submission. We very much appreciate the suggestions and thoughts you have provided.
>
> You are correct to point out that there are some other works exploring inference-time efficiency: it would be more correct of us to write “relatively little” as the number of works is much smaller than that existing for CNNs or language models. We have corrected this point and cited appropriately.
>
>
> We would like to emphasize that a very important quality of our work is that it is inference-time focused, and generalises to unseen graphs. As you mention, there are hardware accelerators for GNNs, but we also note that these are more focused on training-time optimization as they often involve expensive preprocessing steps [1] or require multiple passes to achieve good performance [2]. The work you cite regarding depth compression has also only been shown to work for semi-supervised scenarios, and not the general case. We think that these works are really interesting however they are not able to offer the latency speedups we describe in our work for unseen graphs. Nonetheless, we are confident that combining our approach with other methods can yield even greater results in the future.
>
> In the updated PDF we address your point regarding smartphone deployments by using less absolute language. To expand on our explanation in the paper, there are also concerns about runtime memory usage and data movement. Moving data can be hundreds or thousands of times more expensive energy-wise than an integer add/multiply operation; reducing runtime memory consumption (which we achieve with quantization) is therefore a major driver in reducing overall energy consumption associated with inference [3, 6].
>
> We have incorporated the bold phrase in the Table 1 caption into our abstract.
>
> We have not supplied wallclock training times but instead a table of latency results on CPU and GPU (Table 4); this table is also expanded in Table 10 in the appendix. All the quantization baselines as well as DQ models were implemented using uniform quantization (as it was first mentioned in the Introduction), and therefore all would result in the same speedup during inference. Alternatives to uniform quantization exist [4, 5] and might result in higher speedups. We see this study as an interesting future direction for our work.
>
> We emphasize that one of the benefits of quantization is inference-time improvements once the model is deployed. In this work we did not consider accelerating training using quantization. However, we believe that our insights regarding numerical precision are helpful with regard to using mixed-precision training (e.g. FP16/BF16/TF32) approaches that have become popular for CNNs/Transformer training -- so our work should be useful for achieving training time speedups too.
>
> Finally, we have updated Figure 1 to remove the lines.
>
>
> ______________________________________________________________
> [1] Xiaobing Chen, Yuke Wang, Xinfeng Xie, Xing Hu, Abanti Basak, Ling Liang, Mingyu Yan, Lei Deng, Yufei Ding, Zidong Du, Yunji Chen, and Yuan Xie. Rubik: A hierarchical architecture for efficient graph learning, 2020.
>
> [2] Tong Geng, Ang Li, Runbin Shi, Chunshu Wu, Tianqi Wang, Yanfei Li, Pouya Haghi, Antonino Tumeo, Shuai Che, Steve Reinhardt, and Martin Herbordt.  Awb-gcn:  A graph convolutional network accelerator with runtime workload rebalancing, 2020
>
> [3] Horowitz, M. 1.1 computing’s energy problem (and what we can do about it). In 2014 IEEE International Solid-State Circuits Conference Digest of Technical Papers (ISSCC), pp. 10–14, Feb 2014. doi: 10.1109/ISSCC.2014.6757323.
>
> [4] Sambhav R. Jain, Albert Gural, Michael Wu, and Chris H. Dick. Trained quantization thresholds for accurateand efficient fixed-point inference of deep neural networks, 2020
>
> [5] Raghuraman Krishnamoorthi. Quantizing deep convolutional networks for efficient inference: A whitepaper, 2018.
>
> [6] Hooker S. The Hardware Lottery, 2020: https://arxiv.org/abs/2009.06489

---

### Official Review · AnonReviewer1 · 2020-10-29
**Solid contribution to quantized network training**

**Rating:** 7
**Confidence:** 4

**Review:**

The authors propose a new technique for quantization aware training of neural networks that is specially suited for graph neural networks. They do a good job of motivating the problem by demonstrating that the large variation of input degree in GNNs can lead to unique challenges for numerical precision, forcing a compromise between truncation error and rounding error. Th proposed technique incorporates stochastic masking and quantization proportional to the input degree to allow higher input-degree nodes to operate at higher resolution on average.

The authors demonstrate strong improvements over quantization aware training that treats all nodes equally, achieving relatively small drops in accuracy for a large compression and speedup of GNN inference.

The work is presented in a straightforward and clear manner, with clear applications to important problems.

Two small things that could improve the paper.
* Percentile tracking is a component to the methods, but relies on a reference for full explanation. A more precise statement of this part of the method in the paper itself would help clarify for readers.
* Minor nit, but some acronyms are used before they are defined (such as GCN).

---

> ### Author Response · Authors · 2020-11-16
> **Addressed comments, and other updates incorporated into updated submission**
>
> Thank you for taking the time to review our submission. We very much appreciate your feedback.
>
> We have fixed both your suggestions in the updated PDF we have uploaded. We have also made additional modifications to the document including: expanded the background section presenting QAT in further detail; updated figures 4, 6, 7 and 8 to make them readable to colorblind people interested in our work; updated Figure 1 slightly; and, other changes suggested by reviewers 4 and 5.
>
> Please do let us know if you have any further comments or suggestions.

---

### Official Review · AnonReviewer5 · 2020-11-06
**Official Blind Review #5**

**Rating:** 6
**Confidence:** 2

**Review:**

Let me note that I have very little expertise in quantization and so cannot really judge the significance of such contributions. I am, however very familiar with the GNN literature.

Summary
-------------
A method to train GNNs such that later quantization works well is presented. The authors first analyse the message passing definition to identify those computation steps whose results show the largest variance, and hence suffers most from the imprecision introduced by quantization. Consequently, hey focus on the message aggregation phase of message passing.
They then propose two improvements to more standard quantization-aware training (QAT): (1) applying quantization during the forward pass only on message aggregation outputs (and doing it more often on nodes that receive many messages); and (2) using percentile-based statistics for determining the ranges of values considered during quantization.
Finally, experiments show that the resulting training procedure works well for GNNs on a number of datasets, matching or slightly improving the baseline performances. In most cases, the proposed Degree-Quant method also outperforms baseline QAT methods.

Strong/Weak Points
-------------
* (+) Empirical results show moderate gains over the baseline QAT methods for int8 quantization, and substantial gains for very coarse quantization to int4.
* (+) Thoughtful experimental ablations study the effect of the two improvements separately, and further empirically verify the theoretical analysis of sources of errors.
* (-) The paper is not self-contained and hence not easily readable for people without background knowledge in quantization. While GNNs are fully (though very densely) defined in Sect. 2.1, no technical details on quantization are provided in Sect. 2.2. I ended up skimming some of the cited papers to even understand how values are practically mapped between fp32 and int8. Consequently, Sect. 3.2 is discussing extensions and alternatives to concepts that are simply not explained in the paper.

Recommendation
-------------
I think this paper can be accepted and would be useful for the very narrow segment of people interested and knowledgeable in GNNs and quantization. However, in the current form, it is inaccessible to a wider audience and I believe that it could be significantly improved in that regard.

Questions
-------------
(1) Message aggregation is identified as a key source of quantization error due to the variance in the number of messages. For graph-level tasks (such as MNIST, CIFAR and ZINC), the aggregation of node representations to a graph representation should lead to a similar problem. Do you have deeper analysis on this aspect?

Detail Feedback
-------------
* Sect. 3.1, end: the mixing of GCN and GIN is somehow confusing and it would be wortwhile to restructure this. (i.e., $\mathbf{y}_{GIN}^{(i)}$ is defined before the equation its used in, but $\mathbf{y}_{GCN}^{(i)}$ after, etc.)
* Sect 3.2 / Alg. 1: I found the use of "mask" / "masking" here highly confusing, as I associate it with removing a value (as in masking of loss components, dropout masks, hiding a human face behind a cat mask, ...), but here the semantics is inverted: masks determine which values are "more visible" (by not applying the quantization to them). Unless this term is already in standard related use in the quantization literature, I would strongly recommend to use a different term here (e.g. "preserved", "protected", ...)
* Fig 5/6 are not readable for colorblind people.

---

> ### Author Response · Authors · 2020-11-16
> **Improved readability and making it more self-contained**
>
> Thank you for your detailed comments and suggestions. We appreciate the time you took to review our submission. We are very keen to incorporate your suggestions to ensure our work is readable for a wider audience.
>
> To ensure our paper is more self-contained, we have expanded our background section (2.2) and describe how integer quantization is commonly applied. We have also addressed your more detailed feedback. We appreciate your comment regarding figures and readability by colorblind individuals: we have addressed this by removing conflicting red/green colors from our figures, and we will make sure to bear this in mind for future submissions.
>
> Regarding your suggestion about the use of the term “masking” -- we have followed the convention set by Fan et al. 2020, although we agree that the term “protected” has clearer semantics in this scenario. We have addressed this.
>
> Regarding your question about whether graph level summarization also benefits from Degree-Quant: this is effectively already tested, although we do not explicitly comment upon it. The percentile aspect of our technique is vital to extracting good performance for graph-level summarisation since not all graphs contain the same number of nodes, causing a tailed distribution. The arguments regarding weight update accuracy also still apply, as the $\frac{\partial \mathcal{L}}{\partial \mathbf{h}^{(i)}_{l+1}}$ term should be more accurate compared to when the activations are always quantized before the summarization. We have expanded on this in the appendix.
>
> We have uploaded a new PDF incorporating your feedback.

---

> > ### Comment · AnonReviewer5 · 2020-11-17
> > **Quantization in Graph Representation Computation**
> >
> > Thank you for your quick reply. If I understand you correctly, you are performing the aggregation of node representations at the graph level in the quantized setting in all cases, correct? I was particularly interested to see if using a similar strategy as for message aggregation (stochastically switching to full precision for some graphs during training) would yield any improvements. The results in Fig. 6 indicate that this is clearly a source of error when going from int8 to int4, so this may be another win.
> >
> >
> > Now that I understand quantization slightly better, I have a follow-up question: how are the masks determined for the nQAT model in the experiments? From the comments and the presence of Table 9 in the appendix, I assume that they are independent of node degrees?
> >
> > Overall, I think the paper could be improved by doing a more careful separate analysis of the two proposed techniques. Table 9 is a start towards that, but I believe having two extra rows in Table 2 (for QAT + degree-based mask choice and QAT + percentile tracking) would be quite insightful. Similarly, having a row for nQAT + percentile tracking would help to understand the impact of the GNN-specific contribution of this paper.

---

> > > ### Author Response · Authors · 2020-11-20
> > > **Added diagrams and more results.**
> > >
> > > Thank you for your comments. We have divided our reply into three sections following the order in your comment:
> > >
> > > 1. At each training step, DQ samples a topology-aware N-element mask, where N is the number of nodes in the graph. Nodes with higher degree have a higher chance to be kept at FP32 at any given training iteration. At each node, the aggregation stage uses messages that are quantised depending on the neighbours’ mask value (therefore some of the messages are FP32 and some are INT8/4); then the result of the aggregation gets quantised depending on whether the node should be protected (kept at FP32 or not), which again depends on the DQ mask. We have added a detailed diagram to the appendix to illustrate how DQ quantization (Figure 11) is applied to GCN in comparison to nQAT (shown in Figure 12). We have also shown in a separated diagram (Figure 13) how graph summarization (which we distinguish from node aggregation -- which we focus on in this work) functions with our method; further techniques for graph summarization are interesting but beyond the scope of this work -- which is a foundational work in this area, and hence focuses on a general method useful to a wide variety of practitioners.
> > >
> > > 2. Regarding how masks are generated for nQAT, you are correct when assuming these are independent of node degrees. The stochastic element of nQAT is only applied to the weight quantization stage: weights are shared across all nodes in the graph. At each training stage and each layer, this mask is generated by sampling from a Bernoulli distribution. The ratio of weights to keep at FP32 is a hyperparameter that we fine tuned individually for each dataset-model pair. It is worth emphasizing again that nQAT is not “graph-aware” like our technique: it is, however, a well performing stochastic quantization technique for CNNs/Transformers.
> > >
> > >
> > > 3. Following your suggestion, we have updated Table 9 with a more thorough analysis on the impact of percentiles when excluded from DQ and when used in combination with nQAT. The expanded table shows that the graph-aware DQ masking is often sufficient to achieve good results, but the addition of percentile results in higher accuracies in most cases and improves stability. We observed that with the use of percentiles much fewer hyperparameter samples were needed to achieve top results. When combining percentiles with nQAT, although helpful to improve performance in some cases, is not sufficient to achieve results comparable to those obtained by the full DQ method (see Table 2). In some cases the drop in performance is severe (e.g. ZINC-GIN).
> > >
> > > In summary: the stochastic element of DQ is vital for extracting the best performance. Adding the percentile to nQAT does not yield performance levels obtained by the full DQ method.
> > >
> > > ______________________________________
> > > When running the experiments for Table 9 we realised that the result we had in Table 2 for DQ W4A4 CORA-GAT was not making use of percentiles, therefore leaving out some performance. We have updated table 2 and put the result previously shown there in Table 9. We gain an additional 6.8%.

---

### Comment · ~Haoran_You1 · 2021-01-16
**Will simply changing the computation order works?**

Dear authors,

Thanks for such great work for making the pioneering step toward GCN quantization, I find that your motivation for the proposed two techniques is from the larger variance induced by aggregation.

I am wondering whether simply changing the computation order of aggregation and combination should work, i.e., we first multiply the quantized nodes' features with quantized GCN weights and then do the propagation/aggregation. Mathematically, such computation flow is equivalent to the normal order, but may potentially alleviate the "larger variance problem".

Sincerely looking forward to your reply!

Best regards,
Haoran

---

> ### Author Response · Authors · 2021-01-18
> **Range is still an issue**
>
> Hi Haoran,
>
> Thanks for reaching out. We aren’t completely sure we understand the question, but we think you’re asking about whether factorizing the weight matrix in/out of the sum would help? We believe this alone is not sufficient, since we still run into the “range” issue we describe in the paper (figure 3).
>
> Please feel free to email us too :-)

---

### Comment · ~Bohan_Zhuang1 · 2021-01-19
**Some missing related works**

Dear authors,

There are some literature [1, 2] on stochastic quantisation during training which I think is quite related to your method.  It would be better to cite and discuss with these works to make your interesting work more complete.

References:

[1]:  "Effective Training of Convolutional Neural Networks with Low-bitwidth Weights and Activations",  in Arxiv 2019.

[2]:  “Learning accurate low-bit deep neural networks with stochastic quantization", in BMVC 2017.

Cheers,

Bohan

---

### Decision · Program_Chairs · 2021-01-07
**Final Decision**

**Decision:**

Accept (Poster)

**Comment:**

The paper presents a quantization aware training method for GNNs. The problem is very well motivated, the method is well-executed, and experiments are also well designed. The paper does seem relatively low on technical novelty.

All the reviewers are positive about the paper, and the paper has certainly improved significantly over the rebuttal phase.

So, we would like to see the paper accepted at ICLR.